# High-Energy, Whole-Body Proton Irradiation Differentially Alters Long-Term Brain Pathology and Behavior Dependent on Sex and Alzheimer’s Disease Mutations

**DOI:** 10.3390/ijms24043615

**Published:** 2023-02-10

**Authors:** Robert G. Hinshaw, Maren K. Schroeder, Jason Ciola, Curran Varma, Brianna Colletti, Bin Liu, Grace Geyu Liu, Qiaoqiao Shi, Jacqueline P. Williams, M. Kerry O’Banion, Barbara J. Caldarone, Cynthia A. Lemere

**Affiliations:** 1Department of Neurology, Ann Romney Center for Neurologic Diseases, Brigham and Women’s Hospital, Boston, MA 02115, USA; 2Harvard-MIT Division of Health Sciences and Technology, Massachusetts Institute of Technology, Cambridge, MA 02129, USA; 3Departments of Neurology, Harvard Medical School, Boston, MA 02115, USA; 4Department of Environmental Medicine, University of Rochester Medical Center, Rochester, NY 14642, USA; 5Department of Neuroscience, Del Monte Institute of Neuroscience, University of Rochester Medical Center, Rochester, NY 14642, USA; 6Mouse Behavioral Core, Harvard Medical School, Boston, MA 02115, USA

**Keywords:** radiation, Alzheimer’s disease, neurodegeneration, CNS, space radiation, proton

## Abstract

Whole-body exposure to high-energy particle radiation remains an unmitigated hazard to human health in space. Ongoing experiments at the NASA Space Radiation Laboratory and elsewhere repeatedly show persistent changes in brain function long after exposure to simulations of this unique radiation environment, although, as is also the case with proton radiotherapy sequelae, how this occurs and especially how it interacts with common comorbidities is not well-understood. Here, we report modest differential changes in behavior and brain pathology between male and female Alzheimer’s-like and wildtype littermate mice 7–8 months after exposure to 0, 0.5, or 2 Gy of 1 GeV proton radiation. The mice were examined with a battery of behavior tests and assayed for amyloid beta pathology, synaptic markers, microbleeds, microglial reactivity, and plasma cytokines. In general, the Alzheimer’s model mice were more prone than their wildtype littermates to radiation-induced behavior changes, and hippocampal staining for amyloid beta pathology and microglial activation in these mice revealed a dose-dependent reduction in males but not in females. In summary, radiation-induced, long-term changes in behavior and pathology, although modest, appear specific to both sex and the underlying disease state.

## 1. Introduction

Artemis I’s successful flight around the moon and return to Earth in 2022 marks the first time since Apollo 17, fifty years prior, that we’ve sent a crew-capable vehicle beyond the radiation protection of Earth’s magnetic shield. Artemis II, planned for 2024, will be our first crewed venture beyond low Earth orbit since the Apollo era and will set the stage for an increasing, sustained human presence on the moon and in deep space. Not least among the challenges of this new era in spaceflight is the threat to human health posed by the chronic exposure to the high energy ionizing radiation environment of space. Composed of electromagnetic waves, subatomic particles, and atomic nuclei up to and beyond massive fully ionized iron, space radiation comes primarily from solar particle events travelling outward from the sun and from omnidirectional, extrasolar galactic cosmic rays. Although the two are quite different in predictability, threat, and energy, the particle flux from both radiation sources consists predominantly of protons. The critical challenge facing the radiation biology research community is determining what risks these low dose rate but chronic, high energy particle exposures will pose to human health.

With this chronic dosing environment, attention has turned towards biological targets previously considered relatively radioresistant, like the brain and heart, due to their predominantly nondividing cell populations. We now appreciate that radiation can inflict meaningful functional deficits in these organs as observed in multiple rodent studies [1,2,3,4], but the theoretical framework at the cell biology level linking these difficult-to-study exposures to any specific non-cancer functional outcome remains tenuous at best—especially when considering how they interact with biological variables like sex and disease state. The same unknowns face patients exposed to radiation therapy. While modern targeting technology in conjunction with the advantageous Bragg peak physics of particle radiation provide substantial improvements over gamma radiation in reducing the dose to normal tissue, these normal tissue exposures are still non-negligible. Additionally, modern treatment planning offers little consideration of potential individual variation in susceptibility to normal tissue effects due to factors like disease risk genetics or even biological sex, which are also primary questions for predicting and quantifying individual risk after space radiation exposure [5,6,7]. Thus, these two applications of radiation biology have the potential, despite their differences, to learn from one another—particularly on questions that pose a fundamental challenge to research, such as those pertaining to the long-term effects after exposure that require long and costly study designs.

To this end, we have conducted one of the first long term investigations of the influence of sex and Alzheimer’s disease comorbidity on neurobehavioral and pathological changes after whole body proton exposure in mice. The theoretical basis and animal model evidence for links between radiation damage and neurodegenerative disease are well reviewed in prior reports [8,9,10,11,12]. This report focuses on comparing these new proton data with the existing proton irradiation literature and findings from our past studies of parallel design investigating sex and Alzheimer’s disease interactions with ^56^Fe irradiation [11,12]. In summary, we show that whole body proton irradiation manifests modest and differential effects on behavior and brain pathology at 7–8 months after exposure depending on the sex and Alzheimer’s mutation status of the mouse. These changes also appear to be largely distinct from the changes observed after ^56^Fe exposure in our prior studies.

## 2. Results

To investigate the late effects of whole body, proton irradiation on AD progression and brain function, male (M) and female (F) wildtype (WT) and APPswe/PS1dE9 transgenic (Tg) mice were irradiated with 0 Gy (sham), 0.5 Gy, or 2 Gy of 1 GeV protons at the NASA Space Radiation Laboratory at Brookhaven National Laboratory at four months of age and were evaluated 7–8 months later (Figure 1A). In general, we observed only modest, sex-specific changes in behavior after irradiation, and these were observed predominantly in Tg over WT mice. Proton irradiation slightly reduced Aβ and microglial activation in M Tg but not in F Tg mice. We observed no long-term impact of proton irradiation on microhemorrhages, hippocampal synaptic, and dendritic density, or most of the assayed plasma cytokines.

### 2.1. Survival

We observed no statistically significant radiation effect on survival in any of the four sex/genotype groups as evaluated with the log-rank test (Figure 1B). Comparing sham irradiated groups, F Tg mouse survival trended lower than M Tg (*p* = 0.087) and F WT (*p* = 0.051) survival, which is consistent with our previous work in this genetic model.

### 2.2. Bodyweight

A three-way ANOVA evaluating change in bodyweight from 4 months (time of irradiation) to 11 months (beginning of behavior tests) of age revealed an unexpected highly significant main effect of radiation exposure (*p* = 0.01) and an expected significant effect of sex (*p* = 0.04) (Figure 1C). However, no pairwise comparisons within sex/genotype groups reached significance with Tukey’s family-wise error correction. Evaluating the radiation effect of absolute bodyweight at 11 months within sex/genotype groups revealed a modest yet statistically significant elevation in 0.5 Gy irradiated F Tg mice compared to sham irradiated F Tg mice (Appendix A). Radiation showed no statistically significant effect on absolute bodyweight within other sex/genotype groups. We also observed no radiation effect on overall health as measured by SHIRPA (Appendix A).

### 2.3. Cognitive and Anxiety-like Behaviors

Beginning at 11 months of age (7 months after irradiation), mice were tested with a battery of behavior tests over the course of 1 month, including assessments of spatial memory, fear memory, anxiolytic behavior, and stress-induced behavior. Overall, we observed modest effects of dose, genotype, sex, and their interactions on cognitive behavior outcomes.

#### 2.3.1. Radiation Effects

Tg mice appeared to be more susceptible to radiation-induced cognitive behavior decrements than their WT counterparts. M Tg mice irradiated with 2 Gy of protons showed a statistically significant decrease relative to sham-irradiated controls in the percent time spent in the novel arm of the spatial novelty Y maze (Figure 2C). This suggests impaired spatial memory as mice naturally show a preference for the novel arm of the maze. This group also displayed a statistical trend for an increased percentage of entries into the open arms relative to closed arms of the elevated plus maze, which could be an indication of reduced anxiety (Figure 2F). As prey animals, mice naturally prefer the closed to the open arms of the maze. In contrast, F Tg mice did not show these radiation effects, but rather they displayed a statistically significant elevation in the percent distance traveled in the center of the open field test in the 0.5 Gy irradiated group relative to both the sham group and 2 Gy group (Figure 2G). As with the effect observed in the M Tg 2 Gy group in the elevated plus maze, this is a suggestion of reduced anxiety since mice naturally avoid exposure in the center of the open field. We did not observe any radiation effects in fear memory (Figure 2D) or stress-induced behavior (Figure 2H).

#### 2.3.2. Genotype Effects

With a three-way ANOVA, we observed a statistical trend for a genotype effect in the percentage of novel arm distance of the spatial novelty Y maze (Figure 2B), a trend for a genotype–sex interaction effect in the percent of time spent in the novel arm of the SNYM (Figure 2C) and a statistically significant genotype effect in stress-induced behavior (Figure 2H). In the latter, comparisons between sham-irradiated WT and Tg mice revealed a statistically significant decrease in Tg males and a trending significant decrease in Tg females in the time immobile in the tail suspension test. This suggests less stress-induced behavior in the Tg groups, but this may also be explained by the known hyperactivity phenotype in this transgenic line [11,12].

#### 2.3.3. Sex Effects

We observed significant sex effects with a three-way ANOVA in the contextual fear conditioning test (CFC) (Figure 2D) and in the percent of open arm time in the elevated plus maze (EPM) (Figure 2E). In general, male mice froze more than female mice in the CFC, suggesting a higher baseline fear memory, and female mice spent a higher proportion of time in the open relative to closed arms of the EPM, suggesting less baseline anxiety. The F Tg mice (except for the 2 Gy irradiated group) exhibited much higher individual variation in this latter test than the other sex/genotype groups.

### 2.4. Non-Cognitive Behaviors

Mice were also tested for assessments of locomotor activity, strength, endurance, motor coordination, and motor learning. Overall, we observed modest effects of dose, genotype, sex, and their interactions on these non-cognitive behavior outcomes.

#### 2.4.1. Radiation Effects

As with the cognitive behavior tests, long-term radiation effects on locomotor activity appeared only in Tg mice. We observed a statistically significant, radiation-induced reduction in locomotor activity in 0.5 Gy irradiated F Tg mice relative to sham irradiated controls as measured by the total distance and total arm entries metrics of the spontaneous alternation Y maze (Figure 3C,D). Despite not reaching statistical significance, the F Tg 2 Gy group had lower means than shams in all locomotor readouts and trended towards significance in open field total vertical counts (Figure 3B). Male mice, on the other hand, showed a radiation-induced increase in locomotor activity in the 2 Gy group relative to the 0.5 Gy group that reached significance for Y maze total distance and total arm entries and trended towards significance for open field total distance (Figure 3A,C,D). Neither of these groups were statistically significantly different from the sham irradiated control group by pairwise comparisons. We also observed an interaction effect between dose and genotype in the three-way ANOVA for the Y maze total distance that trended towards statistical significance (Figure 3C).

Unlike in the other behavior metrics, irradiation produced long-term effects only in WT mice for strength, endurance, and motor coordination and learning outcomes. We observed a statistically significant decrease in the grip strength of 2 Gy M WT mice compared to 0.5 Gy M WT mice though neither group was significantly different from the sham control group (Figure 3E). In F WT mice, we observed a statistically significant reduction in endurance by wire hang time in both 0.5 Gy and 2 Gy irradiated groups relative to sham irradiated controls despite extremely high individual variance in the control group. This endurance metric also produced a three-way interaction effect between dose, genotype, and sex by three-way ANOVA that trended towards statistical significance (Figure 3F). In both male and female WT mice, we saw reductions in motor coordination that trended towards statistical significance for the 0.5 Gy irradiated groups relative to sham irradiated controls. Both of these 0.5 Gy irradiated groups also had lower means than their 2 Gy irradiated counterparts, but this only trended towards significance in the WT M group and was not statistically remarkable in the F WT group (Figure 3G). We observed no radiation effect in motor learning with the percent improvement variable in the rotarod task, though the individual variations were quite high relative to the group means (Figure 3H).

#### 2.4.2. Genotype Effects

We observed statistically significant main effects of genotype in the three-way ANOVA for open field total distance and grip strength (Figure 3A,E) and significant interaction effects between genotype and sex for open field total distance and open field vertical counts (Figure 3A,B). WIth pairwise comparisons of sham irradiated groups, we observed an elevation of locomotor activity in F Tg mice compared to F WT mice for both metrics of the open field test: total distance (trend) and total vertical counts (significant) (Figure 3A,B). We also observed statistically significantly less grip strength in sham irradiated F Tg mice relative to their WT counterparts (Figure 3E).

#### 2.4.3. Sex Effects

The main effect of sex by three-way ANOVA reached statistical significance or trended towards significance for every locomotor metric measured: open field total distance, open field vertical counts (trend), Y maze total distance, Y maze total arm entries. In general, female mice appear more active than male mice except in the case of open field vertical counts. We also observed a statistically significant main effect of sex for the grip strength and rotarod motor coordination tests wherein females generally performed better than males, which is likely related to differential bodyweight.

### 2.5. Beta-Amyloid

In addition to behavior outcomes, we also measured the beta-amyloid (Aβ) load in the brains of the Tg mice. Overall, females had more pathology than males at baseline, and radiation only had a long-term effect in males where it modestly reduced Aβ load.

#### 2.5.1. Radiation Effects

We observed significant radiation effects on Aβ load in male but not in female mice. Insoluble Aβ levels from brain homogenates as measured by MSD ELISA showed a statistically significant reduction of Aβ42 in male 2 Gy mice relative to male 0.5 Gy mice. No significant pairwise change was observed in Aβ40. A 2-way ANOVA with relaxed homoscedasticity and normality assumptions revealed a significant main effect of dose for Aβ40, and this main dose effect trends towards significance for Aβ42. However, this effect was not observed in 1-way ANOVA models within each sex, which did not require relaxed assumptions (Figure 4A,B).

Aβ plaques as measured by R1282 immunohistochemistry show a possible radiation-induced reduction in male but not female mice. We observed a trend towards significance for a reduction in R1282 staining in male 2 Gy mice relative to both male 0.5 Gy and sham irradiated mice, and this also manifests as a statistical trend in a 1-way ANOVA within this M Tg group (Figure 4C,D). Denser Aβ aggregates as measured by ThioS dye, however, did not show any radiation dependence (Figure 4E,F).

#### 2.5.2. Sex Effects

Statistically significant main effects of sex were observed in 2-way ANOVAs with relaxed assumptions for all four measures of Aβ load, and pairwise comparisons between sham irradiated groups revealed statistically significantly higher Aβ40 and Aβ42 levels in female brain homogenates than in male brain homogenates, which is an expected phenotype of this transgenic mouse line (Figure 4).

### 2.6. Gliosis and Microhemorrhage

As with Aβ load, we observed a modest decrease in microglial activation measured by TSPO immunostaining in 2 Gy irradiated male mice relative to sham irradiated controls. Similar to Aβ as well, we observed a statistical trend for higher TSPO staining in females than in males. We observed no statistically significant or biologically meaningful changes in hemosiderin staining representing microhemorrhages dependent on radiation dose, genotype, or sex (Figure 5).

### 2.7. Dendrite and Synapse Density

In general, we observed no statistically significant radiation effects on levels of the dendritic marker MAP2 or on levels of pre- and post-synaptic markers SYP and PSD95, respectively, in the CA1 and CA3 regions of the hippocampus. However, a three-way ANOVA for CA1 SYP revealed a trend for a three-way interaction effect between dose, genotype, and sex, and multiple comparisons revealed a trend for more SYP in sham irradiated F Tg mice than in sham irradiated M Tg mice. For the CA3 region, the ANOVA revealed a trend for a main effect of sex, and multiple comparisons showed a trend for more SYP in 0.5 Gy but not 2 Gy irradiated F WT mice compared to their sham irradiated counterparts (Figure 6).

### 2.8. Plasma Cytokines

Measures of plasma cytokines at 8 months after proton irradiation revealed few small changes dependent on radiation dose and sex. Note the magnitudes of these changes are small for meaningful changes in systemic cytokine signaling, and altogether, the data here show a notable lack of long-term radiation effect on systemic cytokine levels. Heteroscedasticity between groups violated strict assumptions of all three-way ANOVAs, so corrections and nonparametric alternatives were applied where necessary when investigating pairwise differences.

#### 2.8.1. Radiation Effects

We observed statistically significant or trending significant effects of dose on the following plasma cytokines: KC-GRO, IL-5, IL-10, IL-12p70, and TNFα. We observed no dose effects on IFN-γ, IL-1β, IL-2, IL-4, or IL-6. Male mice exposed to proton radiation 8 months prior appeared to have a modest reduction in plasma KC-GRO levels (Figure 7I). By three-way ANOVA, we observed a main effect of the dose that trended towards statistical significance and a statistically significant interaction effect between dose and sex. Dunn’s corrected multiple comparisons after the Kruskal–Wallis one-way ANOVA revealed both dose groups were significantly or trended towards being significantly lower than the sham-irradiated group for both M WT and M Tg mice but not any female mice. Beyond KC-GRO, wherein the most consistent dose effect was observed, we also found significant and trending significant main effects of dose on IL-5 and IL-12p70, respectively. A pairwise investigation revealed a trend for a reduction of IL-5 only in F WT mice after 2 Gy compared to 0.5 Gy and a trend for a reduction of IL-12p70 only in M WT mice after 2 Gy compared to sham-irradiated mice (Figure 7E,H). For IL-10, we observed no pairwise differences between groups but noted an overall trend towards significance of an interaction effect between dose and genotype, as the 0.5 Gy irradiated group is slightly lower compared to sham and 2 Gy groups in male and female Tg mice but slightly higher in male and female WT mice (Figure 7G). TNFα showed no change after irradiation by three-way ANOVA, but M Tg mice irradiated with 2 Gy showed a trend towards a statistically significant reduction in the cytokine compared to sham-irradiated controls in a Dunn’s corrected multiple comparison following a one-way Kruskal–Wallis test of the M Tg group (Figure 7J).

#### 2.8.2. Genotype Effects

Little direct genotype dependence was observed with an ANOVA or pairwise comparisons. However, as mentioned above we did detect a trend for an interaction effect between dose and genotype for IL-10 (Figure 7G) and a trend for an elevation of KC-GRO in M WT over M Tg mice (Figure 7I).

#### 2.8.3. Sex Effects

We observed statistically significant or trending significant effects of sex on the following cytokines: IL-1β, IL-2, IL-4, IL-5, and KC-GRO. As with the observed radiation dose effects, these are modest changes and of uncertain biological significance. After investigation for effects by three-way ANOVA, pairwise comparisons by unpaired, two-tailed *t*-tests were made between sham irradiated groups (F WT vs. M WT; F Tg vs. M Tg). Welch’s t test and the nonparametric Mann Whitney U test were used as appropriate for group variance and normality. For overall effects captured in a three-way ANOVA, only IL-5 returned a statistically significant main effect of sex. By pairwise comparisons, we observed a trend for less IL-5 in M WT compared to F WT and a highly significantly less IL-5 in M Tg compared to F Tg (Figure 7E). An analysis of other cytokines revealed no main effects of sex with ANOVA, but with simple pairwise comparisons, we did observe higher levels of IL-1β in male compared to female mice (highly significant for WT and trend for Tg) (Figure 7B). An analysis of IL-2, IL-4, and KC-GRO all revealed a trend towards statistical significance for a difference between male and female WT but not Tg groups (Figure 7C,D,I).

## 3. Discussion

Complementing our prior studies on short- and long-term sex and Alzheimer’s disease interactions with ^56^Fe irradiation, we report here one of the first investigations of long-term sex- and genotype-dependent effects of proton irradiation on the brain. Ultimately, we discovered a modest differential impact of proton irradiation on long-term beta-amyloid pathology between males and females that does not follow the sex differences observed in our prior studies with ^56^Fe irradiation [11,12]. The idea of ion-specific effects as a general phenomenon is well supported in the literature comparing single ion proton exposures to other radiation species [13,14,15,16,17], though this has rarely been investigated in the context of interactions with neurodegenerative disease [18]. The scope of the following discussion is limited to studies using single ion proton exposures at similar or lower doses and investigating endpoints for which we present new data here. It is important to note that this does not include a growing body of work investigating the CNS after mixed particle irradiation, which in the case of galactic cosmic ray simulations are often predominately composed of protons. While important for assessing spaceflight risks, combination exposures often produce effects not seen in single ion exposures of their constituent species, and this superadditive phenomenon is an open area of research.

This work, as with much of the literature surveyed here, is limited by its use of only acute exposures. This logistical concession, along with the reduction to a single charged particle species, makes for an imperfect translation for the space radiation environment, and dose rate effects are beyond the scope of this discussion. It is also important to note that the ultimate neurobehavioral consequences of spaceflight exposures are confounded by non-radiological stressors, such as fluid shifts and atrophy induced by altered gravity, disrupted sleep and circadian rhythm, and the psychological and microbial consequences of isolation, but understanding the interactions between these stressors must include an understanding of the individual components.

As dose rate limits the interpretation of this data for flight relevance, its potential relevance to therapy must account for targeting. While the 0.5 and 2 Gy doses used here are small relative to tumor-targeted doses, they are substantial for whole-body exposures. Despite potential confounds from physiology outside the CNS, we expect this work may serve as an upper bound for functional consequences arising from normal neural tissue exposure prior to Bragg peak energy deposition. In this regard, the experimental model works well as these protons do not lose enough energy traversing mice to have an appreciable change in LET.

### 3.1. Survival and Bodyweight

We observed no impact of radiation on survival, though F Tg mice were more likely to die over the course of the study than their M Tg and F WT counterparts. However, in our long-term, 1 GeV/u ^56^Fe study, we did observe significantly more attrition in 0.5 Gy irradiated F Tg mice than sham irradiated F Tg mice by 8 months after irradiation. Taken together, this suggests a potential three-way interaction between sex, genotype, and radiation particle species or at least a lesser impact of the much lower linear energy transfer (LET) but much higher particle flux, proton dose on overall survival. In addition, unlike our prior ^56^Fe study, we observed a small but statistically significant dose-dependent change in bodyweight with proton irradiated mice gaining more weight than their sham irradiated counterparts over 7 months across all sex and genotype groups. Bodyweight is not commonly reported in the relevant literature, and to our knowledge, this is the first record of a proton radiation effect on bodyweight in this area. Aside from our prior studies, Parihar et al., (2018) reported no significant change in bodyweight at 1 year after 0.05 and 0.30 Gy of 400 MeV/n ^4^He radiation, and Patel et al., (2017) reported no significant bodyweight changes across 0.1–1.0 Gy of 1 GeV protons as well as gamma (1.0–2.5 Gy), ^24^Si, and ^56^Fe (each 0.1–1.0 Gy) irradiated mice at 5 and 9 months after exposure [14,19]. However, the 1 Gy proton irradiated group appeared to have a substantially lower bodyweight than all others at these timepoints. Note the mice from Parihar et al., (2018) were shipped to CA from BNL, whereas the mice in our study travelled a much shorter distance to MA. Seemingly minor logistical variations from study to study such as this may have an unexpectedly large impact on certain outcome measures, which is important to keep in mind for the following discussion. Regardless, these sex and ion dependencies on such macro health parameters like weight and survival are significant. They suggest that sufficient comorbidity can alter radiation sensitivity—or rather that these doses can appreciably exacerbate an underlying disease state. Seemingly contradictory to this idea, however, we observed no elevation in AD-like pathology in the surviving F Tg mice in either this long-term proton study or our prior long-term ^56^Fe study. Reconciling the two observations suggests a progression to death potentiated by, but separate from, Aβ pathology and neuroinflammation.

### 3.2. Behavior

We recorded small but statistically significant long-term changes in behavior after proton irradiation, predominantly in Tg mice, that also manifested differently between males and females. Overall, Alzheimer’s-like Tg mice exhibited more radiation-induced changes in memory, anxiolytic behaviors, and locomotor activity while their wildtype counterparts were more susceptible to radiation induced changes in strength and motor coordination. It is important to emphasize here that the effect sizes relative to the group variance for these behavior endpoints were small, deficits were not consistently found across multiple readouts for a given function (e.g., anxiolytic behavior as measured by open field center distance vs. elevated plus maze open arm entries), and many outcomes showed no response to irradiation. In WT mice 7 months after 0.5 or 2.0 Gy of 1 GeV proton irradiation, we observed no changes in spatial memory, fear memory, anxiolytic behavior, stress-induced behavior, or locomotor activity in either sex, whereas we did see modest sex- and radiation-dependent changes in motor coordination and muscle strength. In contrast, with Tg mice 7 months after 0.5 or 2.0 Gy of 1 GeV proton irradiation, we observed no changes in fear memory, stress-induced behavior, muscle strength, or motor coordination in either sex, but we did observe minor sex- and radiation-dependent changes in spatial memory, anxiolytic behavior, and locomotor activity.

Spatial Memory: There is little corroboration within the existing low-dose proton literature on the dependence of these outcomes on radiation, which, along with the more general field investigating low-dose central nervous system effects, remains predominantly in an early phenomenological observation phase [1]. Of the studies assessing spatial memory and navigation in non-disease model rodents, Shukkit-Hale et al., (2004) (rats; 1.5, 3, and 4 Gy; 250 MeV), Dulcich et al., (2013) (mice; 2 Gy; 150 MeV), and Patel et al., (2017) (mice; 0.1 and 1.0 Gy; 1 GeV), collectively spanning timepoints 1.5 to 9 months after irradiation, also reported no change in spatial learning/memory after proton irradiation [14,20,21]. Counter to this, Kiffer et al., (2020) (mice; 0.5 Gy; 150 MeV, plus 0.1 Gy ^16^O), and Bellone et al., (2015) (mice; 0.5 Gy; 150 MeV), spanning 6 to 9 months after irradiation, reported worsened spatial learning/memory after proton irradiation [22,23]. Bellone et al., (2015) also reported no effect at 3 months after exposure. In the APP/PS1 Tg mouse model of AD (as used in our study), Rudobeck et al., (2017) reported no proton-induced change in spatial memory at 3 and 6 months after 0.1 or 1.0 Gy of 150 MeV protons, which agrees with this study [10]. Note that across these studies, a variety of behavior tests were used to quantify spatial memory, and these tests are not perfectly comparable. Extrapolating from these limited data, this may suggest a U-shaped dose response curve with a trough somewhere between 0.1 and 1 Gy. This may also have an additional dependence on particle energy with higher energies (and thus lower LETs) manifesting effects at higher dose ranges than lower energies. Comparing to ^56^Fe irradiation, we previously reported a trend for a reduction in the spatial memory performance of male Tg mice at 7 months after 0.1 or 0.5 Gy of 1 GeV/u ^56^Fe irradiation as measured by Y maze spontaneous alternation. We did not observe this decrement in the Y maze in this study at the same time point after proton irradiation, but we did see a modest decrement in the male Tg 2 Gy group’s performance in the newly added spatial novelty Y maze test. Patel et al., (2017) reported no change and high variance in spatial memory 9 months after low doses of proton, gamma, ^28^Si, or ^56^Fe irradiation [14]. Further analysis of particle fluence between studies may reveal a more consistent pattern. Alternatively, this may simply be a manifestation of the inherent noisiness and differential sensitivities of behavior testing.

Fear Memory: Fear memory has been less well studied in this proton context. Sweet et al., (2014) reported no effect on contextual or cued fear conditioning in males or females out to 1 year after 0.1–2.0 Gy, 1 GeV proton irradiation [9]. Raber et al., (2016) reported enhanced contextual fear memory 1 month after 0.1–1.0 Gy, 150 MeV protons but cautioned that this is with respect to an abnormal control group, although this enhancement was also observed after ^28^Si radiation in a prior study [13,24]. Owlett et al., (2020) reported in a 3xTg AD mouse model no change in cued fear conditioning in male or female mice at 7 months after 2 Gy of 50–150 MeV proton exposure or after 0.1 or 1.0 Gy of 600 MeV/u ^56^Fe irradiation [18]. We previously reported a deficit in the contextual fear conditioning of male wildtype mice 7 months after 0.1 or 0.5 Gy of 1 GeV/u ^56^Fe irradiation, but we did not observe radiation-induced changes in the contextual fear conditioning of any sex/genotype group in this proton study. As with the following behavior tests with little repetition in the literature, little can be generalized from these findings at this stage.

Anxiolytic Behavior: We assessed anxiolytic behaviors (alternatively interpretable as increased risk-taking behaviors) by quantifying the mouse’s affinity for the open arms of the elevated plus maze and for the center of the open field arena in both this study and our prior long-term ^56^Fe investigation. Of note, our open field test was performed in a smaller and lower light arena than a standard open field test with the goal of minimizing anxiety effects on quantifying locomotor activity. We observed no changes in wildtype mice in either study and small significant or trending differential changes between sexes and ions in transgenic mice, although none appeared in both the elevated plus maze and open field assessments—only one or the other. In general, when there was an effect, radiation tended to increase anxiolytic (decrease anxiety-like) behavior relative to sham irradiated controls. Dulcich et al., (2013) reported a higher level of anxiety behavior by time spent in the dark chambers of an elevated zero maze in sham control female over male mice that was not observed in females 2 months after 2 Gy, 150 MeV proton irradiation [21]. Patel et al., (2017) reported no changes in anxiety behavior by open field at 5 and 9 months after irradiation with 0.1 or 1.0 Gy of 1 GeV protons or 600 MeV/n ^56^Fe, which agrees with the wildtype mice observations in our two long-term studies [14].

Stress-Induced Behavior: Dulcich et al., (2013) also reported worsened stress-induced activity in the tail suspension test at 2 months after irradiation, which, extrapolating from our data showing no effect, may resolve by 7 months post exposure [21]. Alternatively, the 2-3-fold higher LET of 150 MeV protons compared to 1 GeV protons may account for the difference. In our long-term ^56^Fe study, we observed a small reduction in stress-induced (formerly depressive-like) behavior by the tail suspension test only in the 0.1 Gy female Tg group with no effect on the wildtype mice.

Locomotor Activity: Our male Tg mice displayed elevated locomotor activity in both the Y maze and open field tests after 2 Gy proton radiation. However, in these cases, the statistically significant difference is relative to the 0.5 Gy group rather than the 0 Gy control. This radiation-induced increase in locomotor activity was previously observed after 0.1 Gy and 0.5 Gy ^56^Fe radiation but in female Tg and to a lesser extent in female wildtype mice instead of males, which marks yet another difference between our findings with proton and ^56^Fe radiation. Pecaut et al., (2002) reported transient deficits in locomotor activity in 250 MeV, 4 Gy proton-irradiated female rats that somewhat resolved by 3 months after exposure [25]. Dulcich et al., (2013) reported no effect on locomotor activity 2 months after irradiation, and Patel et al., (2017) showed significant decreases in distance traveled in the open field test at 5 and 9 months after proton irradiation but not after ^56^Fe irradiation [14,21].

Motor Coordination: We assessed motor coordination using the rotarod test and observed trends for radiation-induced changes in wildtype but not Tg mice after 0.5 Gy but not 2 Gy. In our long term ^56^Fe study, we observed no changes in wildtype mice and an improvement in rotarod performance in male Tg mice. The same three studies reporting locomotor activity also reported on motor coordination with the same level of disagreement. Pecaut et al., (2002) rats showed rotarod impairment out to 3 months after exposure to 3 and 4 Gy protons, Dulcich et al., (2013) mice showed no change 2 months after 2 Gy proton exposure, and Patel et al., (2017) mice showed a drop in performance at 5 months that persisted but lost statistical significance by 9 months after 0.1 or 1.0 Gy proton irradiation. Patel et al., (2017) mice also showed no change at either timepoint after ^56^Fe irradiation.

Other Behavior Endpoints: Our study adds wire hang and grip strength tests to the literature. Here we observed small changes of uncertain biological significance in wildtype mice but not transgenic mice. Several papers in this field contain uncorroborated behavior endpoints such as acoustic startle, prepulse inhibition, psychomotor vigilance, object in place recognition, operant responding, and conditioned taste aversion that, like our strength/endurance measures, would need repetition before meaningful interpretation [20,26,27,28,29]. While we do interrogate spatial memory as discussed previously, this study and its companion ^56^Fe study unfortunately lack data on novel object recognition, which is one of the most commonly assessed endpoints in this area [13,14,18,22,27,30]. Of those studies reporting novel object recognition after proton exposure collectively ranging from 1 mGy to 2 Gy, 50 MeV to 1 GeV, and 1 to 9 months after irradiation, only Owlett et al., (2020), using 3xTg AD mice, report no radiation effect (at 7 months after exposure of 9-month-old mice). Among these studies, Rabin et al., (2014) reported the largest cohort covering a wide range of doses, energies, and ions including protons and investigated changes at both 5 and 9 months after exposure. While not investigating neurodegenerative disease interactions per se, they note a strong pattern across multiple ions, energies, and doses of radiation induced deficits in object recognition worsening or, in the case of lower doses, newly manifesting with age. Overall, this concordance in object recognition is rare among behavior endpoints, which generally seem to show high sensitivity to specific experimental conditions and little agreement between low LET and high LET exposures.

### 3.3. Alzheimer’s Pathology and Neuroinflammation

The ion dependence also extends to Aβ pathology. Our initial investigation into the short-term responses after ^56^Fe irradiation revealed changes in female but not male transgenic mice at 2 months after irradiation. Specifically, we observed a decrease after both 0.1 and 0.5 Gy of whole brain insoluble Aβx-40 but not Aβx-42, a decrease in hippocampal ThioS staining but not R1282 staining, and a decrease in both ThioS and R1282 staining in the frontal cortex. This effect appeared transient as it was not replicated in our long-term cohort, assessed 8 months after ^56^Fe irradiation, which showed no Aβ changes in females but rather Aβx-40 and hippocampal R1282 staining increases in males. In this study, at 8 months after 0.5 Gy and 2 Gy of proton irradiation, we observed yet another distinct pattern of change. The Aβ pathology of female mice showed no radiation dependence and the Aβ pathology of male mice showed a modest reduction after 2 Gy in whole brain insoluble Aβx-42 but not 40 and by hippocampal R1282 staining but not hippocampal ThioS staining. These differential findings between measures of Aβ suggest a reduction in diffuse plaques (R1282) but not more compact fibrillary structures (ThioS). This reduction was accompanied by a trend towards reduced hippocampal TSPO staining, a marker of microglia activation. Corresponding hippocampal TSPO staining changes were not observed in either of the previous ^56^Fe studies, but we did observe evidence for chronic elevation of neuroinflammation by PET imaging with a TSPO tracer in only the higher dose (0.5 Gy ^56^Fe) male transgenic group at 7.5 months after exposure. In no study did we observe significant changes in microhemorrhages after irradiation. In summary, for both proton and ^56^Fe radiation, we observed long-term effects on Aβ pathology in male, but not female, mice, protons tended to lessen, while ^56^Fe tended to exacerbate, pathology, and overall, 0.5 Gy of 1 GeV/u ^56^Fe irradiation seemed to be a more potent Aβ modifier than 2 Gy of 1 GeV proton irradiation.

To our knowledge, there are only two other studies, one using APP/PS1 mice [10] and the other using 3xTg AD mice [18], reporting on proton effects in Alzheimer’s disease models. Rudobeck et al., (2017) investigated male APP/PS1 mice, the model used in this study, at 9 months after 0.1, 0.5, or 1.0 Gy of whole body 150 MeV proton radiation at 3 months of age. Due to the lower energy, these protons have a roughly 2-3-fold higher LET than the 1 GeV protons used in our study meaning lower particle fluence for a given dose. They report elevated Aβ in the dorsal cortex but not in the hippocampus after 1 Gy but not lower doses as measured by 6E10 IHC staining. Given the small magnitude of the changes we observed after 2 Gy and lack of change after 0.5 Gy, this lack of change in the hippocampus is not particularly surprising. They also observed no change in inflammatory cytokines in brain homogenates. Owlett et al., (2020) reported on male and female mice using the 3xTg AD model (with dramatic sex differences in pathology) exposed at 9 months of age to 2 Gy of 50–150 MeV protons and assessed 7 months later alongside groups exposed to ^28^Si and ^56^Fe radiation. They reported little radiation change in the low pathology males but observed in females a decrease in Aβ plaques by 6E10 staining in the subiculum after 2 Gy proton exposure and notably no effect after exposure to the heavier ions. This change was not reflected in insoluble Aβ40 or Aβ42 measured by ELISA in hippocampal homogenates. Neither was any radiation effect observed on CD68 and IBA-1 in the subiculum or CA1, and little radiation effect was observed on mRNA transcripts for inflammatory proteins. They also observed no radiation-induced change in tau pathology by staining and Western blot with a pT205 antibody.

In addition to these Alzheimer’s disease studies, Sweet et al., (2014) and Raber et al., (2016) also investigate inflammatory responses after proton irradiation, although in wildtype mice instead of in disease model animals [9,13]. Sweet et al., (2014) observed transient elevation in hippocampal GFAP staining 3 months after exposure and a transient decrease in ICAM staining 1 month exposure, both of which resolved by 6 and 3 months, respectively, and remained so out to 12 months. Both effects were observed in 0.1, 0.2, 0.5, 1.0, and 2.0 Gy exposure groups. Raber et al., (2016) observed radiation changes in the cytokine levels of hippocampal homogenates. No changes were observed after proton radiation alone, and little change was seen after ^56^Fe alone, but combination exposure slightly elevated IL-12p70 and TNFα and greatly reduced IFN-γ, MDC, and eotaxin at 1 or 3 months after exposure.

Ultimately, an observation of an interaction between Alzheimer’s disease pathology and particle radiation is still relatively new in the scientific literature. Most reports, some suggesting exacerbation [31] and others suggesting amelioration [32], describe gamma exposures, and it remains an open question whether gamma and various particle radiation species have similar or divergent impacts on neurodegenerative disease progression.

### 3.4. Synaptic and Neuronal Density

We report no changes in immunofluorescent quantification of synaptic and dendritic markers in the CA1 and CA3 hippocampal areas. However, this may only be an issue of methodological sensitivity as several studies have reported alterations in synaptic density and dendrite morphology after similar proton doses and at similar timescales after irradiation. Parihar et al., (2015ab) both reported short-term alterations in dendritic endpoints but use different methods for spine quantification and ultimately disagree on the location and nature of the change. Parihar et al., (2015a) reported compromised dendritic complexity and mushroom spine density by golgi stain of pyramidal neurons with dendritic complexity in the CA1 region worsening 1 month after 2 Gy but not after 0.5 Gy or in the dentate gyrus (DG) after either dose and with mushroom spine count roughly doubling after both doses in CA1 and only after 0.5 Gy in the DG [27]. In contrast, Parihar et al., (2015b) investigated the granule cell layer neurons of the molecular layer of the DG using IMARIS (Bitplane Inc.) dendritic reconstruction and analysis from high resolution immunofluorescent confocal microscopy. They reported reduced SYP, elevated PSD95, fewer immature spines, and severe dendritic complexity reduction—specifically with dendritic thinning—at 1 month after 1 Gy but not 0.1 Gy, reductions in filopodia but not mushroom spines after both doses, and reduced SYP and elevated PSD95 after both doses [33]. Furthermore, Rudobeck et al., (2017) reported the elevation of SYP in whole brain homogenates after irradiation in WT but not in Tg mice, Kiffer et al., (2020) (coupled with ^16^O exposure) reported severe reductions in mushroom spines at 9 months after exposure, and Chmielewski et al., (2016) also reported PSD95 elevation 1 month after irradiation [10,22,34]. Altogether, these marked dendritic and synaptic changes after particle radiation are widely observed phenomena, both in these proton studies and in several other investigations with heavier ions not surveyed here.

## 4. Methods

### 4.1. Mice

All animal use was approved by the Harvard Medical School Office for Research Subject Protection—Harvard Medical Area Standing Committee on Animals and the Brookhaven National Laboratory Institutional Animal Care and Use Committee. Mice were either APPswe/PS1dE9 transgenic (APP/PS1, Tg) or age- and sex-matched C57BL/6J wildtype (WT) littermates. The APPswe/PS1dE9 mice had the Swedish APPK594N/M595L human transgene as well as the PS1dE9 human transgene, both of which are under a mouse prion promoter. These mice developed AD pathology, including extracellular amyloid deposits in the prefrontal cortex and hippocampus by 5–6 months of age, and by 7–8 months of age, they developed further Aβ plaque deposition, microhemorrhages, gliosis, and cognitive deficits [35,36,37]. Mice were irradiated at 4 months of age at the Brookhaven National Laboratory (BNL) in Upton, NY, USA. At 11 months of age, they underwent behavioral testing and were euthanized the following month (12 months of age) by CO_2_ asphyxiation. Mice were housed at a constant temperature on 12 h light/dark cycles with ad libitum access to food (PicoLab Rodent Diet #5053) and water at BWH, HMS, and BNL.

### 4.2. Irradiation

In September 2016, all mice were shipped to BNL, allowed to acclimate for 3–5 days, and transferred to the NASA Space Radiation Laboratory (NSRL). Mice were loaded into ventilated 50 mL conical tubes, which were then loaded into foam holders and carried into the exposure cave. There, the mice received whole-body irradiation delivered laterally. Mice were irradiated in batches of 10 animals. The irradiation consisted of 1 GeV protons (~0.22 keV/μm) at either 0, 0.5, or 2 Gy at a rate of 0.2 Gy/minute. Particle fluence was ~1.4 × 10^9^ ions/cm^2^ for the 0.5 Gy exposures and ~5.6 × 10^9^ ions/cm^2^ for the 2 Gy exposures. This equates to an estimated average of 4400 and 17,600 particle traversals through a 20 μm diameter circular target (roughly one cell body). There were 12–18 mice per sex/genotype/dose group. Following irradiation, mice were returned to their cages. Sham-irradiated mice were put into the 50 mL ventilated conical tubes and loaded into the foam holders for an equivalent time to that of the irradiated mice, but they were not taken into the exposure cave.

### 4.3. Euthanasia and Tissue Preparation

After CO_2_ euthanasia, mice were transcardially perfused with 20 mL of phosphate buffered saline. Brains were extracted and cut in sagittal sections. One hemibrain was snap-frozen in liquid nitrogen and stored at −80 °C, and the other was fixed overnight in 4% paraformaldehyde (PFA), cryoprotected with 10% and 30% sucrose, and embedded in OCT as previously described [8].

### 4.4. Behavioral Tests

At 7 months after irradiation when the mice were 11 months of age, 7–8 mice per group underwent multiple behavioral tests (Table 1). Testing was performed at the Harvard Medical School Mouse Behavior Core Facility. The following behavior tests are presented in the order in which they were performed. Acoustic startle and paired pulse inhibition tests were conducted between TST and CFC, but the data for these particular tests were compromised and have been excluded.

Open Field (OF): The objective of the OF test is to measure locomotor activity, context habituation, and anxiety. Mice were placed in the center of a 27 cm square test chamber and given 1 h to explore. Locomotor activity was quantified using a computer-assisted infrared tracking system, which analyzed the total distance traveled and counts for vertical exploration. Anxiety-like behavior was measured by the percent distance traveled in the center of the field.

Grip Strength: The Grip Strength test measures muscular strength. Mice were placed on the grip strength meter and slowly pulled back by their tail until they released the meter. The force of the pull was measured by a force transducer and recorded. Mice underwent 5 consecutive test trials with 10 s intertrial intervals. The average grip strength was calculated after eliminating the highest and lowest force measurements.

Rotarod: The Rotarod test measures sensorimotor coordination and motor learning. Mice underwent a habituation trial of 5 min at a constant speed of 4 rpm to acclimate mice to the apparatus. Two accelerating test trials (test 1 and test 2) of 3 min each at a speed increasing steadily over 4–40 rpm assessed motor coordination and motor learning. The latency to fall was recorded. Percent improvement was calculated as (Test2-Test1)-1 X 100.

Spontaneous Alternation Y Maze Alternation (YM): The Spontaneous Y Maze Alternation test measures spatial memory and locomotor activity. Mice were placed in the center of the maze and allowed to explore for 6 min. The total distance travelled, and number and sequence of arm entries were recorded. Alternations were consecutive entries to the three arms (i.e., ABC, ACB, BCA, BAC, CAB, or CBA), and the percent alternation (number of alternations × 100/number of arm visits−2) was used as an index of memory performance.

Elevated Plus Maze (EPM): The EPM test measures anxiety and locomotor activity. The maze consists of 4 arms, 2 open and 2 closed, that extend from a central platform. Each arm is approximately 35 cm long and 5 cm wide, and the entire maze is 90 cm above the floor. Mice were placed in the center of the maze facing toward an open arm and were allowed to explore the maze for 5 min. A computer-assisted video tracking system (TopScan Software, Version 1, CleverSys Inc., Reston, VA, USA) recorded the number of entries in the open and closed arms and total time spent in the center of the maze, the open arms, and the closed arms. The number of closed arm entries was used as a measure of locomotor activity, and the percent of open arm entries and percent spent in the open arm was used to measure anxiety-like behavior.

Wire Hang: The Wire Hang test measures muscle strength and endurance in mice. The testing apparatus is a thin wire, secured to posts with tape, stretched horizontally 40-50 cm above a cage filled with bedding. Laminated paper discs were secured to either ends of the wire to ensure that the mice could not climb down the support posts. Mice were placed on top of the wire and then released after all four paws gripped the wire. Mice underwent three consecutive trials of up to 3 min with 30 s intertrial intervals. The latency to fall was recorded and averaged across each trial.

Spatial Novelty Y Maze (SNYM): The SNYM test determines if a mouse can use spatial cues to distinguish between a novel arm and a familiar arm that had been previously visited. Mice underwent a habituation phase of 3 min in which they could explore the maze while the left or right arm was blocked. After a 2-minute intertrial interval, the arm blockade was removed for the test phase and mice were allowed to explore both arms of the maze for 3 min. A computer-assisted video tracking system measured total distance traveled and time spent in the novel arm and familiar arm. Percent time and percent distance in the novel arm were used as a measure of memory performance.

Tail Suspension Test (TST): The TST measures stress-induced behavior. Mice were suspended by their tails for 6 min and a computer-assisted video tracking system determined how much time the mouse spent immobile, which was considered a measure of stress-induced behavior (formerly termed depressive-like behavior). The mouse’s activity was represented as voltage output and the time below a set threshold was an estimate of how much time the mouse spent immobile.

Contextual Fear Conditioning (CFC): The CFC test measures amygdala/hippocampal-dependent learning. CFC was conducted over two consecutive days, with a training phase on day 1 and a test phase on day 2. On day 1, mice were placed in a conditioning chamber and received 2, 2 s, 0.5 mA foot shocks with an intertrial interval of 2 min. On day 2, mice were placed back into the conditional chamber but did not receive a shock. A computer-assisted video tracking (TopScan Software, Version 1, CleverSys Inc., Reston, VA, USA) assessed fear learning by measuring freezing. Percent freezing in the conditioning context on the test day was used as an index of memory.

### 4.5. MSD Multiplex AB Triplex-40/42 Protein ELISA

Brain homogenization and ELISA protocols were performed as previously described in Liu et al., (2019) and Schroeder et al., (2021) [11,12]. Briefly, TPER (Thermo Scientific, Waltham, MA, USA) insoluble fractions were dissolved in guanidine by overnight mixing at 4 °C and were then centrifuged at 175,000× *g* for 60 min at 4 °C. The supernatant was transferred, aliquoted, and stored at −80 °C. Cerebral levels of Aβ-40 and x-42 were measured simultaneously with the Human/Rodent 4G8 Aβ Triplex Ultra-Sensitive Assay (Meso Scale Diagnostics, Rockville, MD, USA).

### 4.6. MSD Cytokine ELISA

The Proinflammatory Panel 1 (mouse) V-PLEX ELISA kit (Meso Scale Diagnostics, K15048D) was used to quantify plasma cytokine levels of IFN-γ, IL-1β, IL-2, IL-4, IL-5, IL-6, IL-10, IL-12p70, KC-GRO, and TNF-α. Plasma samples were diluted 2.4x with diluent provided by the kit, and the ELISA was conducted following manufacturer instructions (Meso Scale Diagnostics, Rockville, MD, USA).

### 4.7. Immunohistochemistry, Histology, and Quantification

Immunohistochemical methods are described in detail in Liu et al., (2019) [11]. Briefly, 20 μm, OCT-embedded, frozen mouse brain sections were immunolabeled using the ABC ELITE method (Vector Laboratories, Burlingame, Calif., USA). The R1282 rabbit polyclonal anti-A antibody (1:1000, a gift from Dr. Dennis Selkoe, Brigham and Women’s Hospital, MA, USA) was used to assess Aβ pathology. One percent aqueous Thioflavin S (Thioflavin S; Sigma-Aldrich, St. Louis, MO, USA) was used to visualize fibrillar amyloid in plaques and blood vessels. Gliosis was assessed using anti-TSPO rabbit monoclonal antibody (an activated microglial marker, 1:1000 Abcam, Waltham, MA, USA). Microhemorrhages were detected using 2% ferrocyanide (Sigma, St. Louis, MO, USA) in 2% hydrochloric acid. Immunoreactivity of R1282, Thioflavin S, and TSPO was quantified with a BIOQUANT image analysis (Nashville, TN, USA). The percent area of R1282 and TSPO staining in the entire hippocampus (HC) was calculated for 2 equidistant sagittal sections 300 μm apart per mouse. The threshold of detection was held constant during analyses. Thioflavin S labeling was averaged by 3 consecutive sections in the middle plane of the hemibrain. Microhemorrhages were counted and averaged over 6 sections (3 consecutive sections, 2 planes) of each mouse. To identify synaptic and dendritic markers, immunofluorescence staining was used to identify the density of pre-synaptic marker synaptophysin (SYP; Sigma-Aldrich, St. Louis, MO, USA), post-synaptic marker PSD95 (MilliporeSigma, Burlington, MA, USA), and dendritic marker MAP2 (MilliporeSigma, Burlington, MA, USA) in the CA1 and CA3 regions of the hippocampus.

### 4.8. Statistics

Data were analyzed as reported in the specific results and figure captions. In general, a three-way ANOVA was used to evaluate the contributions of dose, sex, and genotype variables to the overall variance of each endpoint. Following the ANOVA, a family of pairwise comparisons was made within each sex/genotype group (i.e., 0 Gy to 0.5 Gy, 0.5 Gy to 2 Gy, and 0 Gy to 2 Gy), and the family-wise error rate was controlled with Tukey’s correction except in the case of the behavior results. Sham irradiated groups differing by only one factor (e.g., M WT 0 Gy to M TG 0 Gy but not M WT 0 Gy to F Tg 0 Gy) were compared with two-tailed *t*-tests independent of the ANOVA. Alterations to this general scheme to correct for violations of the assumptions of normality and equal variance were made as noted. All data were displayed as mean ± SEM. StatView (SAS Institute, Cary, NC, USA) and Prism 9 (GraphPad, San Diego, CA, USA) software were used for analysis.

## 5. Conclusions

In summary, we present here a relatively large body of behavior data, as well as Alzheimer’s disease pathology quantification in male and female transgenic and wildtype mice many months after exposure to high energy proton radiation. We show statistically significant evidence for mild radiation-induced sex-specific and genotype-specific changes in behaviors and pathology. The specific changes observed align poorly with our previous studies of parallel design using ^56^Fe radiation, which suggests, as does much of the literature, a qualitatively different macro-scale biological effect between these low and high LET particle species. The interaction of radiation exposure with neurodegenerative disease remains an open question that has thus far only been scantly investigated.

## Figures and Tables

**Figure 1 ijms-24-03615-f001:**
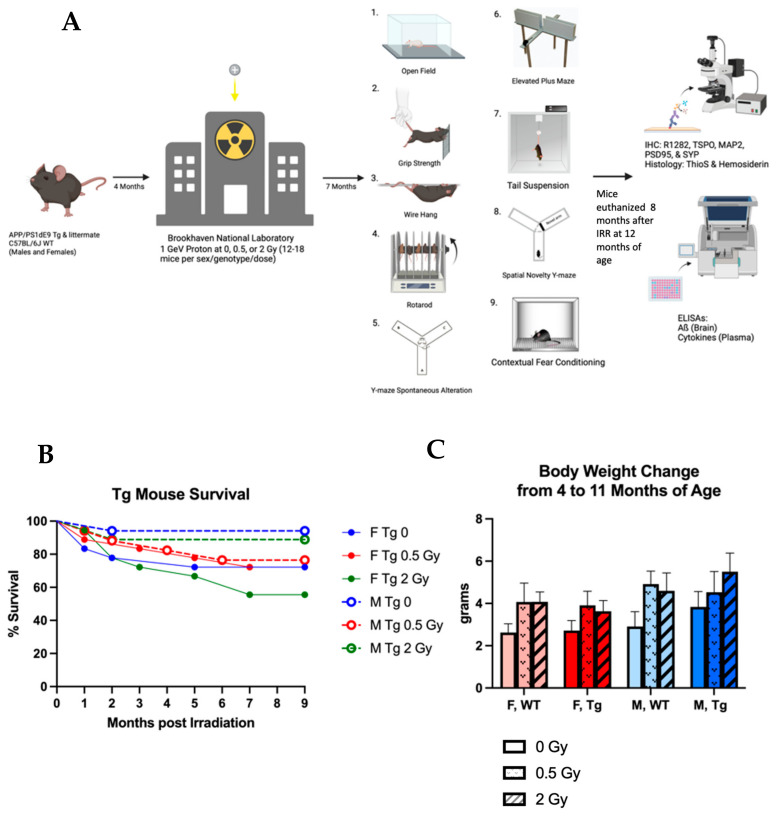
(**A**): Study timeline. APP/PS1 (Tg) and WT mice were transported to and from Brookhaven National Laboratory for irradiation at four months of age with 0, 0.5, or 2 Gy of 1 GeV protons (*n* = 12–18 mice/group). Mice underwent behavioral tests starting at seven months post-irradiation (*n* = 7–8 mice/group). The mice were sacrificed and tissues harvested at 13 months of age (nine months post irradiation) for immunohistochemistry, histology, and ELISAs (*n* = 6–16 mice/group). Created with BioRender.com (**B**): APP/PS1 mouse survival curve. No statistically significant radiation effects were observed by the log-rank test. (**C**): Change in bodyweight from 4 to 11 months of age. Proton irradiation had a significant main effect by 3-way ANOVA (*n* = 10–16 mice/group).

**Figure 2 ijms-24-03615-f002:**
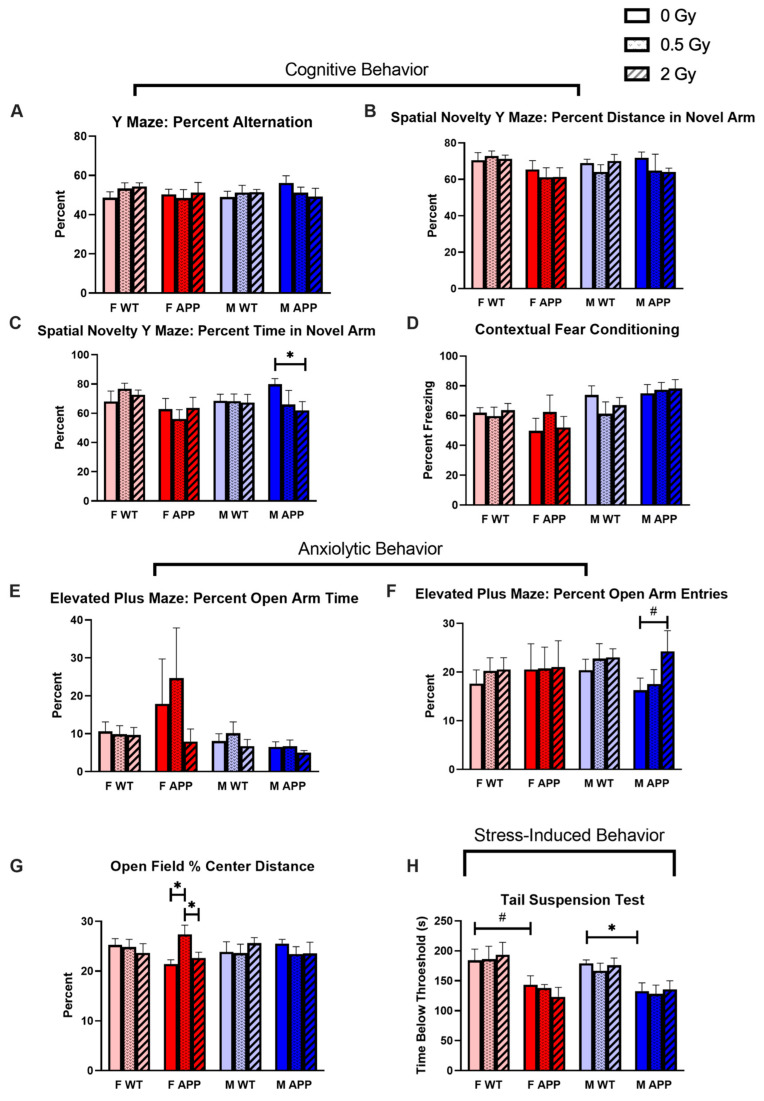
Changes in cognitive behaviors 7 months after a single dose of whole-body proton irradiation depend on sex, genotype, and dose. Four-month-old male and female APP/PS1 (Tg) mice and WT littermates were irradiated with 0, 0.5, or 2 Gy of whole-body proton irradiation at BNL and a subset of these mice (*n* = 7–8 mice/group) went through behavioral tests starting at 7 months post-irradiation. (**A**–**D**): Proton irradiation had no statistically significant effect on memory function by 3-way ANOVA as assayed by the (**A**) Y maze, (**B**,**C**) spatial novelty Y maze, and (**D**) contextual fear conditioning tests. However, multiple comparison testing reveals a significant radiation-induced reduction in spatial novelty Y maze performance in only male Tg mice. (**E**–**G**): Proton irradiation had no statistically significant effects on anxiolytic behavior by 3-way ANOVA. However significant pairwise differences were observed in (**F**) the proportion of open arm to closed arm entries in the elevated plus maze for male TG mice and in (**G**) the proportion of center to peripheral distance travelled in the open field test for female Tg mice. (**H**): Proton irradiation had no effect on stress-induced behavior. #: *p* < 0.1, *: *p* < 0.05. Data were analyzed by 3-way ANOVA followed by 1-way ANOVAs of male and female mice with Fisher’s Least Significant Difference.

**Figure 3 ijms-24-03615-f003:**
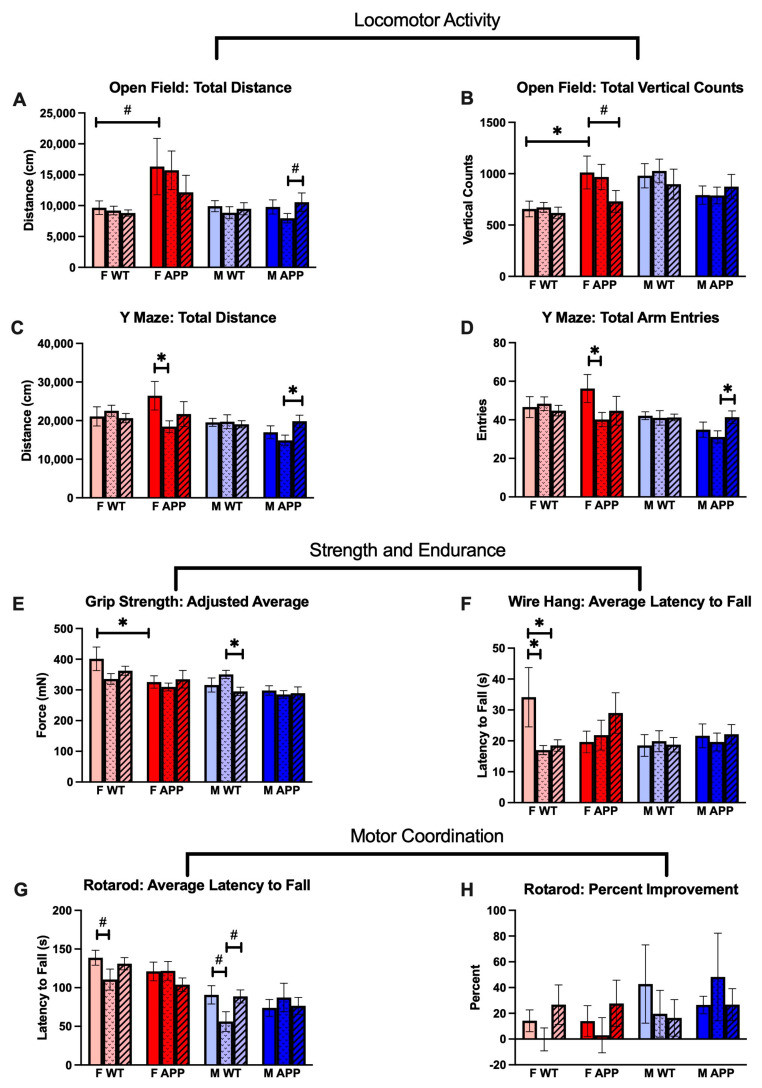
Non-cognitive behavioral changes in APP/PS1 and WT mice 7 months after a single dose of whole-body proton irradiation depend on sex, genotype, and dose. (*n* = 7–8 mice/group) (**A**–**D**): Radiation-induced locomotor activity changes were observed in Tg but not in WT mice. (**E**–**F**): Radiation-induced changes in strength ((**E**), grip strength) and endurance ((**F**), wire hang) manifested in only male and female WT mice respectively. (**G**–**H**): Proton irradiation had late dose-specific effects on (**G**) the motor coordination of wildtype mice of both sexes but not (**H**) motor learning in any group. #: *p* < 0.1, *: *p* < 0.05. Data were analyzed by 3-way ANOVA followed by 1-way ANOVAs of male and female mice with Fisher’s Least Significant Difference.

**Figure 4 ijms-24-03615-f004:**
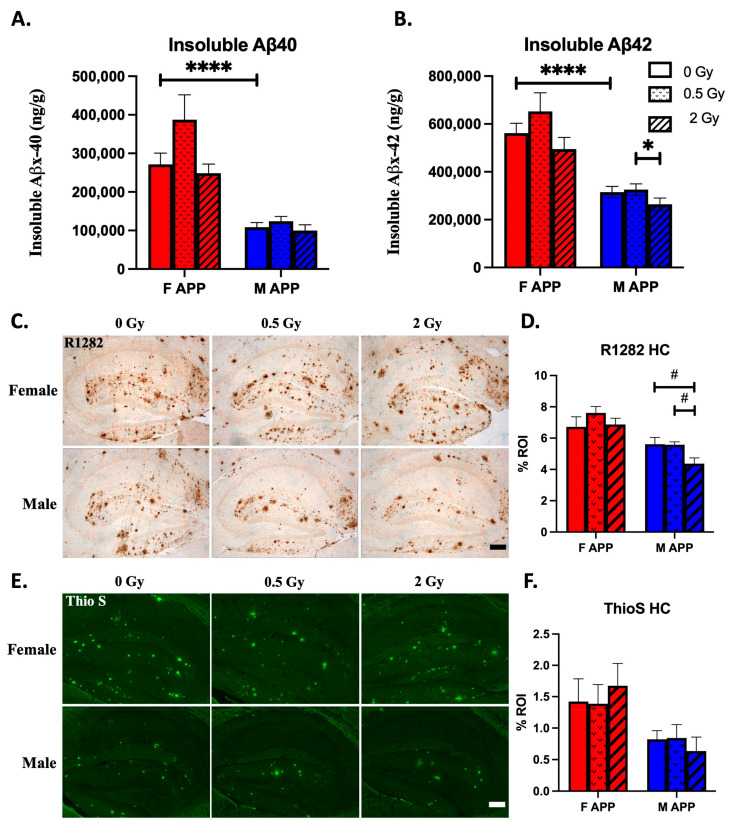
Proton irradiation may have modestly reduced cerebral Aβ levels in Tg male mice, but no change in Tg females was observed. (**A**,**B**): Insoluble Aβx-40 and Aβx-42 levels were quantified by an MSD 4G8 Aβ-triplex ELISA from hemibrain homogenates (*n* = 10–16 mice/group). (**C**,**D**): R1282 immunohistochemical (IHC) staining and quantification of hippocampal beta-amyloid (*n* = 8 mice/group). (**E**,**F**): Thioflavin S staining and quantification of fibrillar protein aggregates in the hippocampus (*n* = 7–8 mice/group). Scale bars: 100 μm. #: *p* < 0.1, *: *p* < 0.05, ****: *p* < 0.0001. Data were analyzed by 1-way ANOVA within sex (after failing parametric assumptions for 2-way ANOVA) followed by post-hoc multiple pairwise comparisons using Tukey’s correction. ELISA quantification of Aβ42 in males violated ANOVA assumptions of normality and was analyzed with a nonparametric Kruskal-Wallis test followed Dunn’s multiple comparisons test. Unirradiated control groups were compared by planned, 2-tailed t-tests independent of ANOVA.

**Figure 5 ijms-24-03615-f005:**
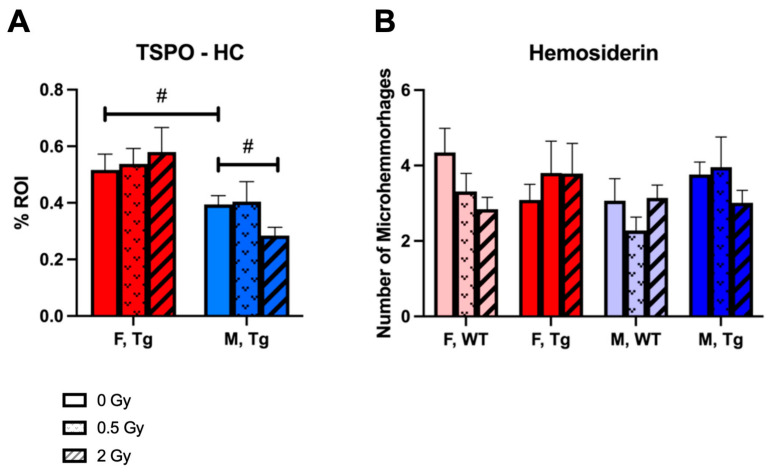
Proton irradiation modestly decreased gliosis in Tg male mice but not in Tg female mice and did not affect the number of microhemorrhages in either sex. (**A**): Immunohistochemical staining of microglial marker TSPO in hippocampus as a measure of gliosis. (**B**): Hemosiderin staining quantified over near-midline sagittal whole brain slice to assess microhemorrhages (*n* = 7–9 mice/group). #: *p* < 0.1. Data were analyzed by 2-way and 3-way ANOVA for TSPO and microhemorrhages respectively followed by post-hoc multiple pairwise comparisons using Tukey’s correction. Unirradiated control groups were compared by planned, 2-tailed *t*-tests independent of ANOVA.

**Figure 6 ijms-24-03615-f006:**
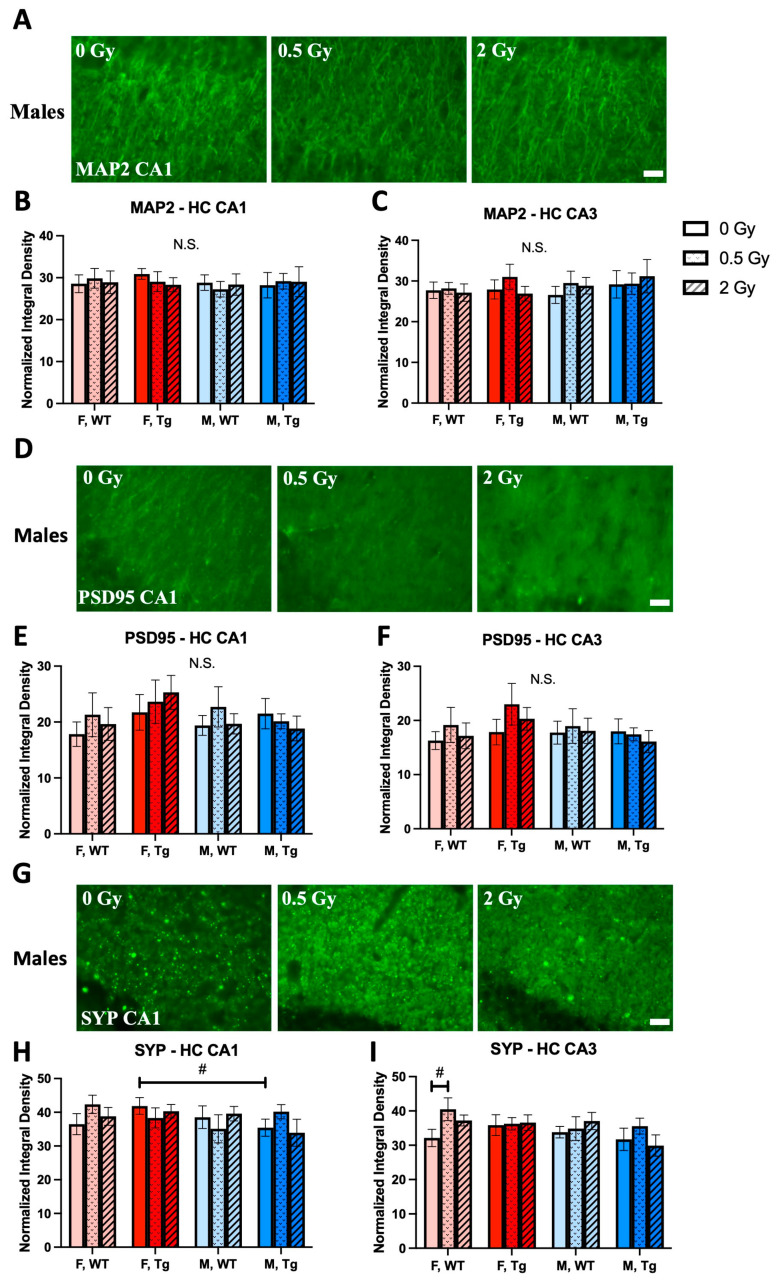
No late effects of proton irradiation on expressions of dendritic marker, (**A**–**C**) MAP2, or synaptic markers, (**D**–**F**) PSD-95 and (**G**–**I**) Synaptophysin in hippocampal CA1 and CA3 regions were observed. Immunofluorescent labeling of synaptophysin, PSD-95, and MAP2 were performed on fixed mouse brain cryosections (*n* = 6–9 mice/group, 2 sections/mouse/marker). Scale bars: 100 μm. #: *p <* 0.1. N.S.: not significant. Data were analyzed by three-way ANOVAs for main and interaction effects followed by 2-way ANOVAs within sex for Tukey’s corrected multiple comparisons between dose groups. Shams were compared across sex/genotype groups by planned, unpaired, 2-tailed t tests independent of ANOVA.

**Figure 7 ijms-24-03615-f007:**
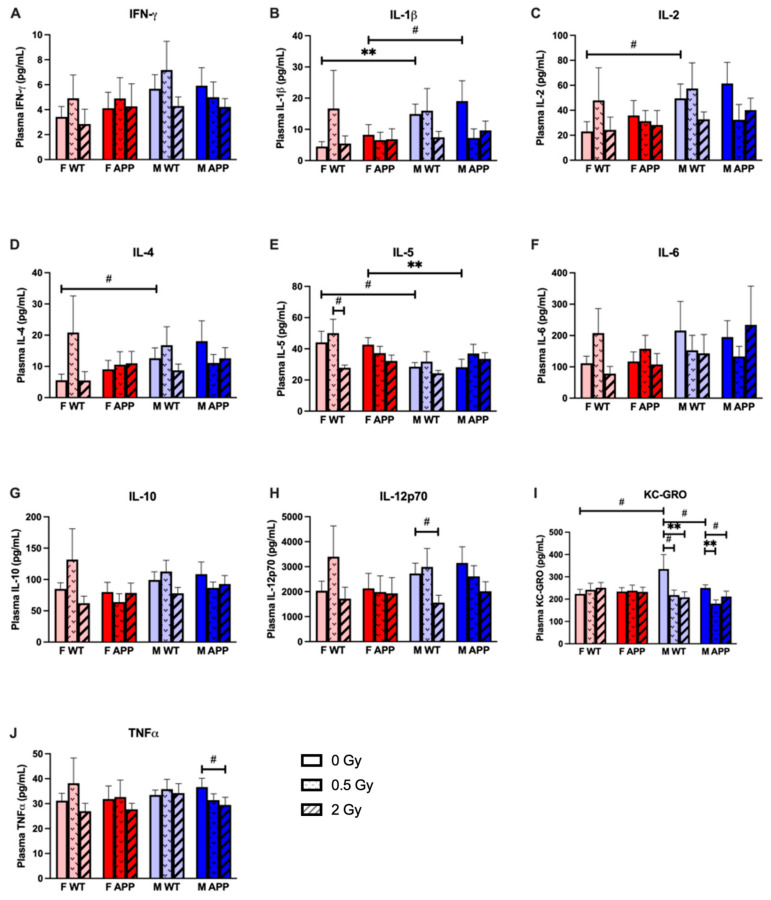
Plasma levels of 10 inflammation-related cytokines 9 months after proton irradiation measured by MSD ELISA (**A**) IL-γ, (**B**) IL-1β, (**C**) IL-2, (**D**) IL-4, (**E**) IL-5, (**F**) IL-6, (**G**) IL-10, (**H**) IL-12p70, (**I**) KC-GRO, and (**J**) TNF-α (*n* = 10–16 mice/group). Modestly lowered levels of KC-GRO were observed in irradiated Tg and WT male mice compared with unirradiated controls. TNFα was also lowered after irradiation in Tg males only. Additionally, WT male mice saw a statistical trend for a decrease in plasma levels of IL-12p70 after 2 Gy of proton irradiation compared to unirradiated controls. #: *p* < 0.1, **: *p* < 0.01. Data were analyzed by ANOVA with unequal variance corrected (Welch’s) and non-parametric (Kruskal-Wallis) versions as required. Unirradiated control groups were compared by planned, 2-tailed *t*-tests independent of ANOVA with unequal variance corrected (Welch’s) and non-parametric (Mann-Whitney) versions as required.

**Table 1 ijms-24-03615-t001:** Behavioral tests. Overview of neurobehavioral functions and their associated test in the test battery. Note, this does not reflect the order in which tests were performed.

Behavioral Measures	Tests
General Health	SHIRPA
Grip and Muscle Strength	Grip Strength (GS) Wire Hanging (WH)
Locomotor Activity	Open Field (OF) Spontaneous Alternation Y Maze (YM) Elevated Plus Maze (EPM)
Motor Coordination	Rotarod
Motor Learning	Rotarod
Anxiety	Open Field (OF) Elevated Plus Maze (EPM)
Depression	Tail Suspension Test (TST)
Spatial Memory	Spontaneous Alternation Y Maze (YM) Spatial Novelty Y Maze (SNYM)
Fear Memory	Contextual Fear Conditioning (CFC)

## Data Availability

All data available upon request to corresponding author.

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
