# Peer review of "High-Energy, Whole-Body Proton Irradiation Differentially Alters Long-Term Brain Pathology and Behavior Dependent on Sex and Alzheimer’s Disease Mutations"

_ijms, 2023, doi:10.3390/ijms24043615_

Round 1

Reviewer 1 Report

Reviewed manuscript investigates long-term changes in behavior and brain pathology between male and female Alzheimer’s-like model and wildtype mice after exposure to high-energy protons. The authors used many behavioral methods to confirm their hypothesis which is valuable for this study. Authors present their data in an understandable and clear manner.

I have only one suggestion for improving the title and manuscript. The animals were irradiated with protons at either 0, 0.5, or 2 Gy. They claimed in the title and entire manuscript that animals were irradiated with low doses. The 0.5 and 2.0 Gy are not low doses for mice! As I know from the literature, the low doses are considered below 100 mGy for acute low-dose exposures as applied in the reviewed manuscript. Therefore, authors should add comments concerning that terminology and probably change that in the title.

Author Response

Thank you very much. We agree that the paper would be improved without the "low dose" language and have removed it throughout. 

Reviewer 2 Report

This paper on “Low-dose, high-energy, whole-body proton irradiation differ-2 entially alters long-term brain pathology and behavior depend-3 ent on sex and Alzheimer’s disease mutations” investigates the effects of high energy particle exposure.

Overall the paper is valuable and informative. The paper is well organized and the data support the evidence. This paper is clear, well written with sound scentific basis and we commend the authors for investigating the influence of sex and Alzheimer’s disease comorbidity on neurobehavioral and pathological changes after low dose, whole body proton exposure

Nevertheless, I think that some statements require further discussion/explanation:

pag 2, lines 56 to 62

While particle radiotherapy, in contrast to space radiation exposure, is 56 delivered focally, acutely, and with orders of magnitude higher doses, the normal tissue 57 surrounding the target volume receives a much lower dose due to the advantages pro-58 vided by modern targeting technologies and the physics of particle radiation. So, while 59 high dose cell killing and tumor control remain the primary goal of cancer treatment, the 60 secondary goal of minimizing surrounding normal tissue damage falls within the suble-61 thal exposure domain that also applies to spaceflight.”.

Please discuss/explain in the light of proton therapy being an enticing treatment modality in radiotherapy largely based on the physical property of the Bragg peak, where the majority of the proton dose is deposited across a very narrow range, with very little to no ‘exit dose’ to normal structures.

pag 9, lines 317-326

Though pairwise comparisons across sex were not assessed, we observed significant 318 sex effects by 3-way ANOVA in the contextual fear conditioning test (CFC) [Figure 2D] 319 and in the percent of open arm time in the elevated plus maze (EPM) [Figure 2E]. In gen-320 eral, though pairwise comparisons between sham-irradiated groups did not reach signif-321 icance, male mice froze more than female mice in the CFC suggesting a higher baseline 322 fear memory, and female mice spent a higher proportion of time in the open relative to 323 closed arms of the EPM suggesting less baseline anxiety. The F Tg mice (except for the 2 324 Gy irradiated group) exhibited much higher individual variation in this latter test than 325 the other sex/genotype groups.”

Please explain as the whole paragraph is confusing.

pag 17, lines 537-540

Furthermore, due to practical limi-537 tations on study designs, the literature surveyed here predominantly investigates the se-538 quelae of acute exposures, which makes for an imperfect translation to the chronic expo-539 sure nature of the space environment.”

Please include an analysis of the limitations.
